# Oligonucleotide Therapeutics for Age-Related Musculoskeletal Disorders: Successes and Challenges

**DOI:** 10.3390/pharmaceutics15010237

**Published:** 2023-01-10

**Authors:** Thomas A. Nicholson, Michael Sagmeister, Susanne N. Wijesinghe, Hussein Farah, Rowan S. Hardy, Simon W. Jones

**Affiliations:** 1MRC Versus Arthritis Centre for Musculoskeletal Ageing Research, Institute of Inflammation and Ageing, University of Birmingham, Birmingham B15 2TT, UK; 2Institute for Metabolism and Systems Research, University of Birmingham, Birmingham B15 2TT, UK

**Keywords:** skeletal muscle, sarcopenia, osteoarthritis, osteoporosis, rheumatoid arthritis, oligonucleotides, cartilage, bone, synovium

## Abstract

Age-related disorders of the musculoskeletal system including sarcopenia, osteoporosis and arthritis represent some of the most common chronic conditions worldwide, for which there remains a great clinical need to develop safer and more efficacious pharmacological treatments. Collectively, these conditions involve multiple tissues, including skeletal muscle, bone, articular cartilage and the synovium within the joint lining. In this review, we discuss the potential for oligonucleotide therapies to combat the unmet clinical need in musculoskeletal disorders by evaluating the successes of oligonucleotides to modify candidate pathological gene targets and cellular processes in relevant tissues and cells of the musculoskeletal system. Further, we discuss the challenges that remain for the clinical development of oligonucleotides therapies for musculoskeletal disorders and evaluate some of the current approaches to overcome these.

## 1. Introduction

Oligonucleotide therapeutics represent a relatively novel class of drug, with the potential to modulate drug targets that were previously considered intractable, and with the benefit of fast clinical development times. In this review, we consider the opportunities for the development of oligonucleotide therapeutics for the treatment of age-related musculoskeletal disorders including osteoarthritis (OA), rheumatoid arthritis (RA), osteoporosis and sarcopenia. With the focus on the tissues of the musculoskeletal system, namely skeletal muscle, cartilage, bone and synovium, we evaluate the successes and challenges of oligonucleotide modalities to modulate target gene expression and ultimately to provide therapeutic efficacy.

### 1.1. Sarcopenia

Sarcopenia, the age-related decline of skeletal muscle mass and strength, is a major contributor to increased frailty, longer hospitalisation and poor healthspan [1,2,3]. Furthermore, since skeletal muscle is the major organ for insulin-mediated glucose uptake, sarcopenia has a profound effect on metabolic health, being a risk factor for the development of metabolic disorders such as type II diabetes [4,5,6,7]. As such, the socioeconomic costs of sarcopenia are significant, with an estimated annual excess cost in the UK of £2.5 billion [8]. 

The drivers of sarcopenia are multifactorial and include inherent biological ageing, obesity [9,10], reduced physical activity [11], and chronic inflammatory conditions such as RA [12], chronic obstructive pulmonary disease (COPD) [13,14], chronic kidney disease (CKD) [15] and advanced liver disease [16]. These sarcopenic drivers ultimately impact one or more processes including the activation of muscle satellite cells and their differentiation into myotubes, termed myogenesis, which is important for repair and remodelling [17,18], innervation and motor unit modelling [19], and the balance between muscle protein synthesis and muscle protein breakdown [11,20,21]. Understanding the key regulator genes that mediate these cellular processes offers the opportunity to identify targets for therapeutic intervention. However, as yet, there are no approved pharmacological therapeutics to combat sarcopenia. Consequently, therapeutic management consists of nutritional interventions and both endurance or resistance exercise regimes [22], which in elderly populations are often of limited efficacy and associated with poor patient compliance.

### 1.2. Osteoporosis

Maintenance of healthy bone requires efficient bone remodelling, a process constantly performed by bone forming osteoblast and bone resorbing osteoclast cells. Disrupting the delicate balance between osteoblast and osteoclast activity in favour of bone resorption can lead to the development of osteopenia and eventually osteoporosis, characterised by a significant reduction in bone mineral density and deterioration of the trabecular bone microarchitecture. Osteoporosis is primarily an age associated disease, with bone mass beginning to decline from around the age of 40 in the absence of loading stimuli such as resistance exercise [23]. Additionally, genetic risk factors include female gender, a family history of osteoporosis and rheumatoid arthritis, while obesity, low calcium and vitamin D intake, therapeutic glucocorticoid use, smoking and chronic alcohol consumption represent significant environmental risk factors [24,25,26,27,28]. Loss of bone mineral density and bone mass in turn significantly increases the incidence of fractures in osteoporotic individuals and ultimately is associated with increased mortality [29,30].

Bisphosphonates are the first line treatment for individuals with osteoporosis, acting to reduce bone resorption through increasing apoptosis of osteoclasts [31]. However, amongst many side effects, chronic treatment with bisphosphates can increase the incidence of osteonecrosis, microfractures and atypical fractures of the femur [31,32]. Consequently, bisphosphonates are often only efficacious for a limited time before treatment must be stopped, at least temporarily [33]. More targeted therapies have since been developed to either prevent osteoclast activation through sequestering RANK ligand (denosumab) an activator of osteoclasts, or by targeting inhibitors of the Wnt signalling pathway of osteoblast formation including sclerostin and Dickkopf-1 (DKK1) [34,35]. However, these treatments are also associated with side effects and are typically far more expensive [32]. This is particularly important as osteoporosis carries a significant economic burden, estimated to be £4 billion per year in the UK [36]. Critically these costs are set to increase substantially as populations continue to age, with the global incidence of hip fractures alone estimated to increase by up to 310% by 2050 [37]. Therefore, novel oligonucleotide therapies could provide a cost effective and targeted approach to treat osteoporosis and osteolysis.

### 1.3. Arthritis 

Arthritis can be broadly classified into two categories, immune-mediated inflammatory arthropathies such as RA, and degenerative arthritis, which is synonymous with OA. In all cases, patients experience joint pain and joint stiffness, leading to progressive joint degeneration and disability [38]. Globally, these conditions accounted for approximately 20 million cases of RA and over 500 million cases of OA [39,40], posing a considerable socioeconomic burden by impacting healthcare costs and reducing working life expectancy.

As a systemic chronic autoimmune disease, RA is characterised by inflammation of the synovium joint lining, termed synovitis, which occurs predominantly in the small joints [41]. The expansion of synovial fibroblasts and their interaction with macrophage-like synoviocytes drives the recruitment of pro-inflammatory leukocyte populations and promotes an inflammatory joint environment. This pro-inflammatory state promotes catabolic pathways and suppresses anabolic pathways in articular cartilage cells (chondrocytes), skeletal muscle cells (myoblasts/myotubes) and subchondral bone osteoblasts [42]. Furthermore, the inflammatory hyperactivation of osteoclasts, which resorb bone, facilitates local bone destruction, pannus formation and systemic osteoporosis [43]. Together, this results in the loss of articular joint cartilage, skeletal muscle atrophy and bone resorption, ultimately driving joint destruction and increasing frailty. This, in turn, is frequently exacerbated in RA patients receiving therapeutic glucocorticoids to manage disease activity, where steroids such as prednisolone further promote systemic muscle wasting and osteoporosis [28,44].

Importantly, despite historically being seen as solely a wear and tear degenerative disease of the cartilage, it is now widely accepted that OA is a disease of the whole joint, encompassing the synovium, subchondral bone, skeletal muscle as well as the cartilage [45]. Although not an overtly inflammatory condition, synovitis is present early in the disease course of OA, as detected by histology, MRI and ultrasound [46,47,48]. OA synovial fibroblasts adopt an inflammatory phenotype, particularly in obese patients [49,50], and express and secrete increased levels of pro-inflammatory cytokines such as TNFα, IL-6 and IL-1β. These cytokines drive cartilage matrix degeneration by promoting the expression and release matrix metalloproteases (MMPs) and aggrecanases from the chondrocytes, which mediate Type II collagen and aggregan proteoglycan degradation, respectively [51,52]. Subchondral bone in OA is referred to as sclerotic, with bone marrow lesions and trabecular thickening [53] due to the dysregulation activity of the osteoblasts and osteoclasts [54], eventually resulting in the formation of osteophytes. Furthermore, skeletal muscle weakness is a common feature of OA in both the knee and hip [55] and may accelerate disease progression. The drivers of this muscle weakness are not clear, but type II atrophy has been reported in the quadriceps of patients with knee OA, and in patients with non-painful OA, suggesting it may not be solely due to reduced physical activity [56].

The treat-to-target therapeutic management of RA patients aims to reduce pain and improve joint function by targeting immune activity and inflammation utilising corticosteroids, non-steroidal anti-inflammatory drugs (NSAIDs), disease-modifying anti-rheumatic drugs (DMARDs) and biologics [57]. However, many of these immunomodulatory drugs do not modify disease pathogenesis, and long-term treatment can result in adverse side effects and increased risk of infections. In OA, therapeutic treatment is particularly limited, with no disease-modifying drugs [45,58] and only generic treatments for alleviating pain and inflammation such as NSAIDs, which have limited analgesic efficacy. OA patients can be offered intra-articular injections of corticosteroids to alleviate joint pain. However, the benefit is short-lived and repeated injections can accelerate joint damage [59]. Ultimately, therefore, many OA patients will undergo joint replacement surgery, where a high proportion of patients report poor post-operative outcomes [60]. Certainly, in arthritic joint conditions, there is still a significant unmet need for more targeted and more effective disease modifying therapies.

## 2. Oligonucleotides to Target the Cells and Tissues of the Musculoskeletal System

Successes in the use of oligonucleotides to modulate relevant gene targets and cellular processes in vitro, as well as those oligonucleotides that have demonstrated in vivo efficacy for the major tissues of the musculoskeletal system, including skeletal muscle, bone, cartilage and synovium (Figure 1) are detailed in Table 1 and discussed below.

### 2.1. Skeletal Muscle

The most investigated target for oligonucleotide therapeutics against sarcopenia is myostatin (also known as growth and differentiation factor-8, GDF-8), which acts as a negative regulator of muscle growth in an endocrine and autocrine manner [61]. An early proof-of principle study by Liu et al. demonstrated, in 2008, that either oral or intravenous (i.v) administration (5 mg/kg for 4 weeks) of a myostatin-targeted 2′-O-methyl antisense RNA increased muscle mass in normal mice and in a cancer cachexia mouse model [62]. Repeated systemic administration of myostatin-targeted oligonucleotides appears more effective than one-off intramuscular injection. In a study on normal mice by Kang et al., a single intramuscular injection of 2′O-methyl phosphorothioate RNA (3 nmol) induced transcriptional changes without achieving a significant change in tibialis anterior muscle mass [63]. In contrast, five weekly i.v injections of an octa-guanidine morpholino oligomer (6 mg/kg) achieved a significant increase in soleus muscle mass and myofiber size. These studies established proof-of-principle for myostatin-targeted oligonucleotide therapeutics to increase muscle mass in vivo. 

A range of knock-down strategies and delivery technologies have been tested for myostatin-targeted oligonucleotides. Antisense RNAs targeting the start site, termination site or inducing exon skipping have proven efficacy to increase muscle mass in normal mice [62,63]. Additional targeting strategies have been investigated in vitro. An antisense oligonucleotide against a splicing enhancer sequence within exon 1 reduced myostatin protein levels and enhanced proliferation of human myoblasts [64]. Additionally, small interfering RNAs, complimentary to a promoter-associated transcript, induced transcriptional gene silencing in two mouse muscle cell lines [65]. Further studies have explored chemical modifications and alternative delivery approaches. Myostatin-targeting oligonucleotides conjugated to cholesterol or octa-guanidine dendrimer (also known as vivo-morpholino structures) have demonstrated efficacy to suppress myostatin in skeletal muscle and increase muscle size following i.v administration in mice [63,66]. Another approach involving a nanoparticle complex containing myostatin-siRNA (1 nmol) and atelocollagen, administered by a single intramuscular injection, successfully induced muscle hypertrophy in a caveolin-3 transgenic mouse model [67]. Lastly, intramuscular administration of myostatin-targeted siRNA (2 nM) in combination with muscle-specific microRNAs-1 and −206 (20 nM) was superior compared to either agent alone for promoting muscle regeneration in a rat model of chemically injured tibialis anterior muscle [68]. In summary, myostatin-targeted oligonucleotides have exhibited knock-down efficacy and protective effects against muscle atrophy for a range of targeted gene regions, chemical conjugations and enhanced delivery approaches. 

A few studies have explored oligonucleotides targeting other mediators in the myostatin signalling pathway. Liu et al. showed that targeting Foxo-1 with 2′-O-methyl antisense RNAs (100 µg i.v twice-weekly for 4 weeks) effectively suppressed myostatin, induced the myogenic factor MyoD and increased skeletal muscle mass in normal and cancer-bearing cachectic mice [69]. Investigating down-stream effectors of the myostatin signalling pathway, Ding et al. showed that siRNA-mediated knock-down of p38β mitogen-activated protein kinase (p38βMAPK) reversed catabolic effects in C2C12 cells [70]. However, an oligonucleotide-based knock-down strategy was not tested in vivo in this study, which relied on a muscle-specific gene deletion mouse model for in vivo validation instead. Finally, Pasteuning-Vuhman et al. tested a Vivo-Morpholino antisense oligonucleotide targeting the myostatin/activin type 1 receptor (ALK4) and found mixed results [71]. While muscle regeneration was stimulated in mouse models, concomitant changes in metabolic function and protein turnover resulted in an overall reduction in muscle mass and muscle fibre size. This finding illustrates some of the intricate complexities that pose challenges to developing treatments for sarcopenia. Taken together, ample evidence from rodent models highlights the potential of oligonucleotide-based therapeutics against myostatin as a treatment strategy against sarcopenia. However, to date, clinical trials testing antibody- or protein-based approaches for targeting myostatin signalling for sarcopenia have given mixed results [61]. 

Senescence can deplete the pool of precursor muscle satellite cells that are essential for the maintenance and regeneration of muscle mass [72]. Price et al. have conducted an interesting study demonstrating that siRNA-mediated knock-down of either Jak2 or Stat3 stimulates satellite cell function [73]. Jak2 or Stat3 knockdown rescued the proliferation defects of satellite cells from muscle of aged adult mice. Furthermore, ex vivo transfection of satellite cells from 3-month-old mice enhanced engraftment when these cells were transplanted into cardiotoxin-injured tibialis anterior muscle. In a different study, Pucci et al. used siRNA to target clusterin, a glycoprotein involved in proliferation and cell death [74]. Ex vivo knock-down of clusterin restored proliferative capability and improved markers of tissue damage repair in human myoblasts from elderly patients with osteoporosis. Lastly, in vitro siRNA-mediated knockdown of cyclin-dependent kinase inhibitor 1 (p21) prevented induction of senescence markers in primary mouse muscle cell cultures treated with serum from CKD patients [75]. These examples illustrate the potential of oligonucleotide therapeutics to reverse senescence changes in muscle precursor cells, thereby promoting maintenance and regeneration of healthy skeletal muscle. Further validation of findings from in vitro studies is needed to evaluate the therapeutic potential of this approach in animal models and ultimately in clinical studies. 

The balance between protein synthesis and degradation is disturbed in sarcopenia, with suppression of the anabolic IGF-1/insulin-pAkt-mTOR signalling pathway and activation of the catabolic ubiquitin-proteosome system [76]. The latter commonly involves activation of specific E3-ubiquitin ligases marking muscular proteins for degradation, namely MuRF1 (aka Trim63) and atrogin-1 (aka MAFbx). Sun et al. report that siRNA-mediated suppression of tumour necrosis factor receptor-associated factor 6 (TRAF6) attenuated dexamethasone-induced muscle atrophy in mice, with concomitant reduction in MuRF1 and atrogin-1 expression [77]. The siRNA was administered by intramuscular injection to tibialis anterior muscle at a dose of 5 nmol every 3 days for 2 weeks. Further in vivo studies of oligonucleotides targeting muscle protein turnover are sparse. Nevertheless, numerous studies have employed siRNA knock-down strategies to explore regulatory components of protein synthesis and degradation in rodent-derived muscle cell lines [78,79,80,81,82,83,84,85,86]. It remains to be explored whether these targets can be exploited for oligonucleotide-based therapies in vivo. 

### 2.2. Bone

One mechanism to reduce bone resorption is to prevent osteoclast formation by inhibition of NF-κB (a positive regulator of osteoclastogenesis) in myeloid lineage cells present within the bone marrow [87]. Shimizu et al. utilised decoy oligonucleotides (0.25–1 μM, 1–7 days) to inhibit NF-κB in rabbit bone marrow mononuclear cells [88]. Such oligonucleotide treatment prevented osteoclastogenesis in response to vitamin D3 treatment. Osteoclast function was also demonstrated to be decreased in oligonucleotide treated cells at 10 days. To validate these findings in vivo, the authors utilised ovariectomised rats, an osteoporotic model mediated by loss of oestrogen production. Two weeks of oligonucleotide therapy delivered to the tibia and femur by mini osmotic pumps (30 μg/kg/h), significantly decreased osteoclast activity in osteoporotic rats, whilst increasing bone calcium and mass [88]. In a similar study, Sato et al. also targeted NF-κB to reduce osteolysis mediated by the accumulation of wear particles following joint replacement therapy [89]. Treatment with an NF-κB decoy oligonucleotide in mice via subcutaneous injection (5 μM, every other day for 2 weeks) was again beneficial, increasing bone mineral density relative to PBS treated controls and preventing particle induced bone resorption through inhibition of osteoclast activation and differentiation. Oligonucleotide treated animals also demonstrated a favourable cytokine profile with increased IL-1 receptor antagonist and osteoprotegerin (OPG), with a reduction in osteoclastogenesis-associated proteins including TNFα and RANK ligand [89]. Research by Dong et al. has since corroborated these findings, demonstrating that even a single dose (5–10 μg) of an antisense oligonucleotide targeting TNFα, an activator of NF-κB signalling, could similarly reduce metal particle mediated osteolysis [90]. In a following study, the authors also demonstrated the potential to use a biodegradable cationic hydrogel as an effective delivery system for such oligonucleotides (5% of the 20 μg load demonstrated to be delivered after 72 h), to ensure localised and sustained delivery. Indeed, osteoclast number and bone resorption were still significantly reduced following 2 months of treatment [91].

Osteoporosis and osteolysis have also been demonstrated to occur secondary to cancer, particularly following metastasis to bone [92,93]. One mechanism identified to drive such bone loss is the upregulation of matrix metalloprotease 13 (MMP13) at the bone-tumour interface [94]. In a comprehensive study, Nannuru et al. provide evidence that antisense oligonucleotides targeting MMP13 (delivered by intraperitoneal injection, 50 mg/kg/day) were remarkably able to prevent any substantial tumour driven bone loss in mice, again through reducing osteoclast activity [94]. Since MMPs have also been implicated in non-cancer driven osteoporosis [95,96], development of similar oligonucleotide therapies targeting other MMPs may also be effective in such patients.

Chronic glucocorticoid treatment can also drive osteoporosis, via an upregulation of osteoclastogenesis, coupled with increased osteoblast apoptosis and impaired osteoblast differentiation and dysfunction [97,98,99]. Antisense oligonucleotides targeting cannabinoid receptor 1 (CB1) were demonstrated to be effective in the prevention of glucocorticoid mediated bone loss when administered concurrently to rats weekly, for 5 weeks (5–20 μg/kg/day) [100]. The authors suggest a potential mechanism for this positive effect was due to prevention of glucocorticoid-induced accumulation of bone marrow adipocytes [100]. 

In contrast to inhibiting osteoclast function and bone degradation as discussed so far, the alternative approach to increase bone mass via promoting an increase in bone synthesis. DKK1 is a negative regulator of bone formation, acting to inhibit the Wnt receptor and the low-density lipoprotein receptor-related protein 5 (LRP5) co-receptor complex formation of the canonical Wnt signalling pathway [101]. Consequently, recent attempts have been made to target DKK1 with monoclonal antibodies to increase bone anabolism [35]. However, Wang et al. similarly show that antisense oligonucleotides targeting DKK1 (intraperitoneal injection, 20 μg/kg/day) may prevent reductions in bone mineral density in osteoporotic ovariectomised rats [102]. In addition, trabeculae number were also increased compared to osteoporotic rats, while osteoclast numbers were reduced, ultimately increasing bone strength [102].

Finally, in addition to oligonucleotide mediated modulation of cellular function, a novel use of oligonucleotides in bone biology could be to deliver bone promoting factors to the site of implantation during joint replacement therapy to prevent aseptic loosening, a major complication of such procedures. To this end, Wolfle et al. coated oligonucleotides onto the surface of titanium implant material. These oligonucleotides had a complementary sequence to other oligonucleotides conjugated to BMP2 and VEGF, thus enabling localised delivery of these factors to the native bone microenvironment upon implantation (2 ng BMP2/mm^2^ or 2 ng VEGF/mm^2^ were immobilized on the orthopaedic implant surface) [103]. Utilising this model, the authors demonstrate increased bone formation, angiogenesis and critically increased incorporation of titanium implant material to the tibia, 12 weeks post-surgery in osteoporotic rats [103].

Collectively, there is evidence that oligonucleotide targeted therapy may offer a novel therapeutic approach to both prevent bone breakdown through inhibition of osteoclast activity and increase bone mass through increasing bone formation in osteoblasts. Continued development of such oligonucleotide treatments could provide valuable tools to maintain bone mass in individuals at risk of osteoporosis and osteolysis. However, it is evident that studies utilising primary human bone cells and progress to clinical trials are now required.

### 2.3. Articular Cartilage

Degeneration of the articular cartilage is a characteristic feature of arthritis, most notably in OA. Articular cartilage, unlike most tissues within the human body, is devoid of blood vessels, nerves and lymphatics. Instead, it is made of dense layers of extracellular matrix (collagens, proteoglycans, non-collagenous proteins and glycoproteins) which in healthy tissue is maintained by specialised cells known as articular chondrocytes. This proteoglycan rich matrix is hydrated with water, providing the cartilage with its load-absorbing properties. In OA, chondrocytes adopt a hypertrophic phenotype, and cartilage homeostasis is disturbed by increased production of MMPs such as MMP3 and MMP13 and aggrecanases ADAMTS 4 and ADAMTS5 [104], which degrade type II collagen and aggrecan proteoglycans, respectively [51,52]. In both RA and OA, the expression of cartilage proteases is exacerbated by the increased presence of inflammatory cytokines in the synovial joint fluid. Furthermore, chondrocytes are also capable of propagating these inflammatory changes through secretion of IL6 which forms a IL6/sIL6r complex within the synovial fluid, thereby binding to and trans-activating membrane bound gp130, propagating IL6 secretion [49]. Thus, the development of therapeutics that are capable of targeting the inflammatory-meditated catabolic pathways in the articular chondrocytes represents a disease-modifying therapeutic strategy in arthritis. 

In vitro studies using human chondrocytes have shown that stromal cell-derived factor (SDF)-1 mediated MMP13 induction can be reduced either using siRNA targeting the SDF-1 receptor CXCR4, or using antisense oligonucleotides (ASO) targeting transcription factors c-Fos and c-Jun to inhibit MMP-13 promoter activity [105]. Furthermore, with the advances in next generation sequencing, a number of changes in the non-coding transcriptome have been found to be altered in OA cartilage, providing a plethora of targets for oligonucleotide antisense therapeutics that would be deemed intractable with small molecule or antibody-based drugs. To this end, human chondrocytes treated with ASO against miR-320a reduced IL-1β-mediated release of MMP13 [106], whilst miRNA oligonucleotide mimics of miR-98 and miR-146 resulted in a decrease in TNF-α and MMP13 production [52], demonstrating the critical role miRNAs play in modulating the inflammatory and degradative phenotype of chondrocytes.

In vivo, several candidate oligonucleotide therapies designed to reduce cartilage degeneration have been studied in preclinical animal models of arthritis. In an anterior cruciate ligament (ACL) rodent model, inhibiting miR-29b-3p using an intra-articular injection of miR-29b-3p antagomir (800 pmol) ameliorated chondrocytes apoptosis and cartilage loss through targeting of PGRN [107]. Similarly, in a surgically induced rodent model of OA, knockdown of miR-3680–3p (which is elevated in OA cartilage) by intra-articular injection of miR-3680–3p antagomir (50 nM, twice a week) alleviated cartilage destruction [108]. Overall, these studies provide promise that oligonucleotide therapies could be developed that effectively target and modulate the activity of pathological and aberrant proteins and miRNAs, which mediate cartilage degradation in arthritic diseases.

### 2.4. Synovium

In arthritis, the synovium undergoes a number of pathological changes including increased proliferation of resident synovial fibroblasts, angiogenesis and vascularisation and increased infiltration of inflammatory immune cells, which drive a chronic inflammatory state within the synovial joint termed synovitis [42]. Pro-inflammatory cytokines such as TNFα, IL-1β and IL6, secreted by activated inflammatory synovial fibroblasts, are a major source of the inflammatory milieu in synovitis [109], and are central drivers of both articular cartilage degradation in OA through induction of MMPs and aggrecanases [105,110], and bone resorption in RA via the induction of RANK Ligand and subsequent activation of osteoclasts. Furthermore, synovitis is associated with greater pain severity in patients with knee OA [111,112], with distinct synovial fibroblast subsets identified in knee OA patients that promote neuronal growth and survival [112]. Indeed, it is known that several pro-inflammatory cytokines present in arthritis synovial fluid can sensitise afferent joint nociceptors and thus increase pain perception [113,114]. Therefore, designing effective therapeutics that can specifically target inflammatory synovial fibroblasts is a strategy for both slowing progression of joint damage and for the alleviation of joint pain.

In vitro studies have demonstrated the efficacy of antisense oligonucleotides in modulating the inflammatory synovial fibroblast phenotype. In one such study, a locked nucleic acid (LNA) antisense targeting the long non-coding RNA MALAT1, which is highly expressed in obese inflammatory OA synovial fibroblasts, resulted in reduced expression of CXCL8 and reduced proliferation [50]. Similarly, the proliferative and inflammatory phenotype of synovial fibroblasts has been modulated by oligonucleotides targeting key metabolic enzymes that have been shown to be dysregulated in the activated fibroblast phenotype [115]. An siRNA targeting the glutamine metabolic enzyme, glutaminase 1 (GLS1), inhibited the proliferation of RA synovial fibroblasts [116], whilst in OA synovial fibroblasts the expression of IL6 was reduced [117]. It has also been demonstrated that it is possible to modulate the inflammatory phenotype of synovial fibroblasts through targeting the obesity associated adipokine, leptin. Leptin is elevated in the synovial fluid of RA and OA patients [49], and modulating leptin signalling through inhibition using an antisense oligonucleotide targeting the long leptin receptor isoform (OBRI) reduced leptin-mediated CXCL8 secretion and IL6 expression in OA synovial fibroblasts [118,119]. Similarly, siRNA targeted inhibition of insulin receptor substrate (IRS-1) and reduced leptin-induced IL-6 production in OA synovial fibroblasts [119]. Another approach to modulating the inflammatory synovial fibroblast phenotype is through targeting central inflammatory pathways that mediate the synthesis and secretion of prostaglandins, which are important mediators of inflammation. To this end, antisense oligonucleotides targeting the NF-κB p65 subunit were found to reduce IL1β-mediated COX2 protein expression in RA synovial fibroblasts [120]. 

An alternative approach to combating synovitis is to target the hyperplastic nature of arthritic synovial fibroblasts. To accomplish that, inhibition of the anti-apoptotic protein FLICE-inhibitory protein (FLIP) using a phosphorothioate-modified oligodeoxynucleotide (5 µM for 24 h) was found to sensitize RA synovial fibroblasts to Fas-mediated apoptosis [121]. Similarly, siRNA mediated silencing of the immunomodulatory lectin, Galectin-9, in RA synovial fibroblasts increased their apoptosis [122], whilst antisense oligonucleotides targeting Notch-1 protein reduced the proliferation of TNFα stimulated fibroblasts [123]. 

To date, despite the plethora of in vitro studies that have demonstrated the efficacy of oligonucleotides to target and modulate the activated synovial fibroblast pathotype, very few oligonucleotide therapeutics have transitioned into demonstrating efficacy in targeting the synovium in preclinical in vivo models. Of those that have, research conducted by Weng et al., targeted DKK1, the Wnt signalling pathway inhibitor, in a surgically induced experimental model of OA [124]. Intraperitoneal administration of a 21-mer end-capped phosphorothioate antisense against DKK1, at a dose of 20 µg/kg/week for up to 12 weeks, resulted in reduced synovial vascularity [124]. Similarly, siRNA targeting FoxC1, which in vitro were found to reduce the proliferation and secretion of pro-inflammatory cytokines in OA synovial fibroblasts, when delivered by intra-articular injection (2 nmol biweekly for 8 weeks) in an in vivo DMM mice model reduced development of synovitis (with a thinner synovium that exhibited reduced immune cell infiltration), as well as reduced cartilage degeneration [125]. Collectively, these data support the paradigm that oligonucleotide therapies targeting the synovium can modify disease progression in arthritis.

**Table 1 pharmaceutics-15-00237-t001:** Oligonucleotide targeting in muscle, bone, synovium, and cartilage tissue.

Tissue	Target	Oligonucleotide	Study Model	Administration	Function	Reference
Muscle	atrogin-1 and MuRF1	siRNA	In vitro, L6 myotubes	-	Combined atrogin-1 and MuRF1 knock-down prevents dexamethasone-induced myotube atrophy	[79]
	PPARβ/δ	siRNA	In vitro, L6 myotubes	-	PPARβ/δ knock-down reduces FOXO1 and MuRF1 expression and protein degradation in dexamethasone-treated myotubes	[78]
	p38β MAPK	siRNA	In vitro, C2C12	-	MAPK knock-down prevents activin A induced catabolic activity	[70]
	TFEB	siRNA	In vitro, C2C12	-	TFEB knock-down prevented angiotensin II-induced MuRF1 expression and atrophy	[80]
	C/EBPβ	siRNA	In vitro, L6 myoblasts and myotubes	-	C/EBPβ knock-down inhibited dexamethasone-induced increase in protein degradation, atrogin-1 expression and muscle cell atrophy	[81]
	p21	siRNA	In vitro, primary mouse satellite cells	-	p21 knock-down restored proliferative capacity of uraemic serum-treated muscle progenitor cells	[75]
	FOXO3	siRNA	In vitro, C2C12	-	FOXO3 knock-down prevented iron-induced upregulation of atrogin-1 and MuRF1 and reduction in myotube diameter	[82]
	myostatin	antisense oligonucleotide	In vivo, MF1 or C57Bl10 mice	IM (ineffective); IV, weekly for 5 weeks (effective)	repeated intravenous administration of myostatin targeting vivo-PMO induced soleus muscle hypertrophy	[63]
	myostatin	siRNA	In vivo, caveolin-3 transgenic mouse dystrophy model	IM, single injection	nanoparticle complex with myostatin-siRNA increased muscle size and fibre size	[67]
	myostatin	cholesterol-conjugated siRNA	In vivo, CD-1 mouse	IV, single injection	myostatin knock-down increased skeletal muscle mass and strength	[66]
	Myostatin, miR-1, miR-206	siRNA and/or miRNA	In vivo, Lewis rat, barium chloride-induced muscle injury	IM, single injection	combined administration of myostatin-siRNA and microRNAs improves in situ skeletal muscle regeneration	[68]
	myostatin	2′-O-methyl antisense RNA	In vivo, BALB/c mouse, cancer cachexia model	oral or IV, twice weekly for 4 weeks	myostatin knock-down induced skeletal muscle hypertrophy	[62]
	Foxo-1	2′-O-methyl antisense RNA	In vivo, BALB/c mouse, cancer cachexia model	IV, twice weekly for 4 weeks	Foxo1 knock-down suppressed myostatin and induced MyoD and skeletal muscle hypertrophy	[69]
	myostatin	antisense chimeras	In vitro, human primary myoblasts	-	antisense oligonucleotide-mediated myostatin knock-down enhanced myoblast proliferation	[64]
	FOXO4	siRNA	In vitro, C2C12	-	FOXO4 knock-down reversed myotube atrophy from siRNA-mediated WNK1 silencing	[83]
	ALK4	antisense oligonucleotide	In vivo, C57BL/6Jico and *mdx* dystrophy model mouse	IM, daily injections for 2 days	Alk4 knock-down stimulated muscle regeneration, but decreased muscle mass	[71]
	Jak2, Stat3	siRNA	In vivo, Pax7-ZsGreen mice, mdx mice, and SV129 mice	IM, single injection of siRNA-transfected satellite cells	Knock-down of Jak2 and/or stat3 enhances regenerative capacity of muscle pre-cursor cells	[73]
			In vitro, aged mouse satellite cells			
	Clusterin	siRNA	In vitro, human primary myoblasts	-	Clusterin knock-down enhanced proliferation in myoblasts from older patients with osteoporosis, but in myoblasts from younger patients with osteoarthritis	[74]
	myostatin	siRNA	In vitro, differentiated mouse primary muscle cells, C2C12 and H2K mdx tsA58 cells	-	Description of epigenetic myostatin silencing, no functional readouts	[65]
	TRAF6	siRNA	In vivo, ICR mice dexamethasone-induced muscle atrophy	IM, every 3 days for 2 weeks	TRAF6 knock-down attenuated dexamethasone-induced muscle atrophy and reduced expression of atrogin-1 and MuRF1	[77]
	PAI-1	siRNA	In vitro, C2C12	-	PAI-1 knock-down prevents dexamethasone-induced atrogin 1 and MuRF1 upregulation	[84]
	SIRP-a	siRNA	In vitro, C2C12	-	SIRPα knock-down promoted insulin signal transduction, pAkt and suppressed protein degradation	[85]
	GSK3β	siRNA	In vitro, C2C12	-	GSK3β knock-down prevented atrogin-1/MuRF1 upregulation and myosin loss with IGF-1 blockade or dexamethasone	[86]
Bone	DKK1	Phosphorothioate antisense oligonucleotide	In vivo, ovariectomized rats. In vitro, primary rat osteoclasts	IP, for 4 weeks (5 consecutive days per week)	Alleviated loss of bone mass and biomechanical property, reduced osteoclast formation	[102]
	NFKB	Decoy oligodeoxynucleotide	In vivo, ovariectomized rat and rats with osteogenic disorder Shionogi. In vitro, rabbit/rat osteoclasts	oral and implantation	Inhibited the formation and activity of osteoclasts, Increased bone length and bone mineral density	[88]
	NFKB	Decoy oligodeoxynucleotide	In vivo, C57BL/J6 mice osteolysis model	SQ, every other day for 14 days	Mitigated the expression of TNF-α, RANKL, and induced the expression of Il-1 receptor antagonist and OPG	[89]
	TNF-alpha	Antisense oligonucleotide	In vivo, C57BL/J6 mice osteolysis model	SQ, single injection	90% of metal particulates induced osteoclastogenesis was suppressed by oligonucleotide delivery	[90]
	TNF-alpha	Phosphorothioate antisense oligonucleotides	In vivo, mouse model of edotoxin-induced bone resorption. In vitro, raw 264.7 cells	implantation	Oligonucleotide delivery via a bio-degradable cationic hydrogel suppressed the expression of TNF-α and osteoclastogenesis	[91]
	MMP13	Antisense oligonucleotide	In vivo, mouse model of tumour induced osteolysis	IP, daily for 5 days, 2 days off, then daily for an additional 4 days	Significantly reduced bone destruction and the number of activated osteoclasts	[94]
	Cannabinoid receptor 1	Phosphorothioate antisense oligonucleotide	In vivo, rats with glucocorticoid induced osteoporosis. In vitro, primary rat osteoblasts	IP, for 5 weeks	Attenuated the deleterious effects of glucocorticoid treatment on bone mineral density, trabecular microarchitecture, and mechanical properties	[100]
	BMP-2 and VEGF-A	Antisense oligonucleotides	In vivo ovariectomized rats	implantation	Significantly improved implant integration with local bone	[103]
Synovium	DKK1	Phosphorothioate antisense oligonucleotide	In vivo, SD rats ACLT OA model. In vitro, primary human SF	IP, weekly for 2, 6 or 12 weeks	Reduced proteinases and angiogenic factors, vessel distribution and formation and cartilage injury	[124]
	FoxC1	siRNA	In vivo, DMM mouse. In vitro, primary human SF	IA, twice weekly for 8 weeks	Inhibited IL-6, IL-8, TNF, ADAMTS-5, fibronectin, MMP3 and MMP13 and proliferation of OA SF and prevented OA development in vivo	[125]
	Notch-1	Antisense oligonucleotide	In vitro, primary human synoviocytes	-	Partially blocked proliferation of RA synoviocytes and inhibited TNFα-induced proliferation in both OA and RA synoviocytes	[123]
	MALAT1	Locked nucleic acid	In vitro, primary human SF	-	Inhibited proliferative and inflammatory phenotype of obese OA SF, reduced CXCL8 expression and secretion and increased expression of TRIM6, IL7R, HIST1H1C and MAML3	[50]
	GLS1	siRNA	In vitro, primary human SF	-	Reduced IL6 expression in OA SF	[117]
	Galectin-9	siRNA	In vitro, primary human SF	-	Increased apoptosis of human RA SF	[122]
	OBRI	Phosphorothioate antisense oligonucleotide	In vitro, primary human SF	-	Reduced leptin-mediated IL-8 secretion via JAK2/STAT3 pathway, whilst activating the IRS1/PI3K/Akt/NF-kappaB-dependent pathway and recruitment of p300	[118]
	IRS1	Phosphorothioate antisense oligonucleotide	In vitro, primary human SF	-	Leptin mediated by inflammatory OA fibroblast phenotype was inhibited resulting in reduced IL-6 via IRS-1/PI3K/Akt/ AP-1 pathway	[119]
	FLIP	Phosphorothioate antisense oligonucleotide	In vitro, primary human synoviocytes	-	Sensitized RA FLS to increased fas-mediated apoptosis by 3-fold	[121]
	p65	Phosphorothioate antisense oligonucleotide	In vitro, primary human synoviocytes	-	Decreased binding of NF-KB to COX-2 promoter and COX-2 protein expression	[120]
Cartilage	c-Fos	Phosphorothioate antisense oligonucleotide	In vitro, primary human chondrocytes	-	Silenced the potentiating action of SDF-1α on MMP-13 promoter activity	[105]
	c-Jun	Phosphorothioate antisense oligonucleotide	In vitro, primary human chondrocytes	-	Silenced the potentiating action of SDF-1α on MMP-13 promoter activity	[105]
	mir-146	pre-miRNA mimics	In vitro, primary human chondrocytes	-	Overexpression significantly attenuated IL-1β induced reduced TNFα production.	[52]
	miR-320a	antisense oligonucleotides	In vitro, primary human chondrocytes	-	Reduced IL-1β mediated release of MMP13 and sGAG whilst enhancing expression of COL2A1 and ACAN	[106]
	miR-98	pre-miRNA mimics	In vitro, primary human chondrocytes	-	Overexpression significantly attenuated IL-1β induced reduced TNFα production	[52]
	miR-29b-3p	antagomir	In vivo, OA rat model, SD rats ACLT/PCLT. In vitro, primary SD rat chondrocytes and SW1353 cells	-	Blocked PGRN, induced apoptosis, inhibited proliferation, and scratch wound closure of chondrocytes. In vivo, ameliorated articular chondrocytes apoptosis and cartilage loss	[107]
	miR-3680–3p	antagomir	In vivo, DMM rat OA model. In vitro, primary human chondrocytes	-	Reversed IL-1β induced chondrocyte injury and delayed OA progression via targeting OGG1. In vivo, attenuated cartilage destruction and loss of cartilage	[108]

Abbreviations: ACLT—anterior cruciate ligament transection, DMM—destabilization of the medial meniscus, IA—intra-articular, IM—intramuscular, IP—intraperitoneal, IV—intravenous, OA—os-teo-arthritis, PCLT—posterior cruciate ligament transection, RA—rheumatoid arthritis, SD—Spra-gue-Dawley, SF—synovial fibroblasts, SQ—subcutaneous.

## 3. Challenges for Clinical Development

Oligonucleotides have huge therapeutic potential for treatment and management across many conditions. Their capacity to selectively bind both by complementary base pairing to target sequences as well as through 3D secondary structures to specific proteins are largely exploited for gene silencing [126]. Currently, there are a total of 11 oligonucleotide-based FDA approved drugs for clinical use in the liver and eye, with three under review targeting skeletal muscle [127]. Advances in both cell biology and nucleic acid chemistry have facilitated improved drug properties, yet many hurdles remain with delivery to target tissues particularly in the musculoskeletal system.

### 3.1. Stability

Stability of oligonucleotides once administered is a particular limiter in clinical translation and depends on protecting nucleic acids from nuclease degradation using chemical modifications and conjugates of peptides, lipids and polymers [128]. First generation antisense oligonucleotides with phosphate backbones had improved circulating lifespan at the consequence of target affinity and immunogenicity, whilst uncharged phosphorodiamidate oligomers had improved target affinity but were inefficient for in vivo delivery [129]. Second generation chemistries improved both binding affinity and resistance to nuclease degradation with ribose modifications. Whilst third generation modifications such as LNAs or cholesterol and GalNac-conjugated oligonucleotides extensively altered the oligonucleotide chemistry, improving both stability and target affinity [127]. Some of these backbone chemistry examples have also been applied to siRNA therapeutics to improve metabolic stability and target binding [130]. Despite these advances, the most appropriate chemical modifications must be determined based on target tissue and modality.

### 3.2. Tissue Penetration and Targeted Cellular Uptake

Another major obstacle in oligonucleotide therapeutics is efficient delivery to target tissues. Due to their hydrophobic nature, oligonucleotides do not pass through the plasma membrane [128]. Thus, systemic and locally administered oligonucleotides must overcome a multitude of biological barriers including nuclear degradation, renal and reticuloendothelial clearance, and uptake by target cells whilst avoiding lysosomal degradation or exocytosis [126,131]. For the development of oligonucleotide therapeutics for musculoskeletal disorders, tissue penetration is a particular challenge. For example, the negatively charged and avascular nature of cartilage prevents deeper uptake of drugs into the cartilage tissue. Furthermore, since the chondrocyte cells, which are responsible for mediating cartilage integrity, are embedded within a matrix, oligonucleotide drugs need to be able to penetrate the full depth of the cartilage tissue [39].

Interestingly, the delivery of FDA-approved oligonucleotide therapeutics largely depends on chemical modifications for tissue delivery as discussed earlier. To this end, second-generation gapmer antisense oligonucleotides have been extensively modified to facilitate delivery to a variety of tissues without a delivery agent [132]. Other strategies are also being explored to improve distribution and pharmacokinetics, including oligonucleotide conjugation to cationic cell-penetrating peptides (CPPs) [133,134], lipid nanoparticles (LNPs) [135,136,137,138], or antibodies [139]. CPPs bypass cell-specific receptors mechanisms, but have limited compatibility with uncharged peptide nucleic acids (PNA) and phosphate morpholino oligonucleotide (PMO) [127]. In contrast, LNPs work to encapsulate oligonucleotides and therefore are compatible with all oligonucleotide types. However, lipid-based strategies such as LNP and exosomes come with their own challenges of costs and scalability [127,131]. Notably, studies have demonstrated the utility of nanoparticles to deliver antisense oligonucleotides into cartilage tissue in vivo, with chondroprotective effects recently reported in models of OA. Sachetti et al. demonstrated that cationic polyethlene glycol (PEG)-modified carbon nanoparticles delivered by intra-articular injection in a surgical murine model of OA (DMM) could penetrate chondrocytes in the superficial zone of cartilage [140]. Whereas, aptamer-decorated PEGylated polyamidoamine nanoparticles were recently used to deliver miRNAs by intra-articular injection into mice, which were retained in the joint space and found to be chondroprotective [141].

More recently, antibody conjugates, which have had success with the targeted delivery of cytotoxic drugs (ADCs), are now also being considered for the targeted delivery of oligonucleotides, termed antibody oligonucleotide conjugates (AOCs) [139]. Such ADCs rely on the binding to cell surface receptors and then subsequent receptor internalisation, thus selection of the receptor target is likely to be critical for achieving sufficient uptake and therefore efficacy. In vitro efficacy of AOCs have been reported in a number of cell types, with gene silencing at between 24–72 h. Xia et al. demonstrated gene silencing at 48 h using the human insulin receptor to deliver an siRNA into HEK293 cells, a human kidney cell line [142]. In vivo, an AOC targeting the TENB2 with siRNA targeting PPIB gene to target prostate cancer cells was dosed 24 mg/kg three times over 3 days by i.v administration into a mouse tumour model, and demonstrated around a 30% knockdown in expression of PPIB [143]. These limited studies hold great promise for the potential for AOCs. However, significant studies are required to determine the optimal conjugation of antibodies to oligonucleotides and to screen receptor targets to identify those receptors specific to the tissue and cell type of interest that will provide the optimal uptake and efficacy, whilst avoiding cytotoxic effects.

## 4. Conclusions

Great strides have been made in modifying oligonucleotides to improve their stability in vivo. However, for age-related musculoskeletal disorders, challenges remain in improving the ability of oligonucleotides to effectively penetrate the tissues of the musculoskeletal system. Furthermore, the selective targeting of specific cell types or, with the advent of single cell data, the desire to target specific cell subsets represents a significant hurdle at present. Overcoming these hurdles is currently the focus of much research and will likely be met via the development of novel bioconjugation strategies. 

Clearly, with the many advantages offered by oligonucleotide therapeutics including their rapid development time, encouraging safety profiles, intrinsic mode of action in targeting genes prior to their translation into proteins, and recent approvals of oligonucleotide-based drugs in other therapy areas, this investment is warranted. Perhaps most importantly though, within the unmet clinical need of musculoskeletal disorders they open the door to an array of previously intractable targets, thus offering new and exciting therapeutic strategies.

## Figures and Tables

**Figure 1 pharmaceutics-15-00237-f001:**
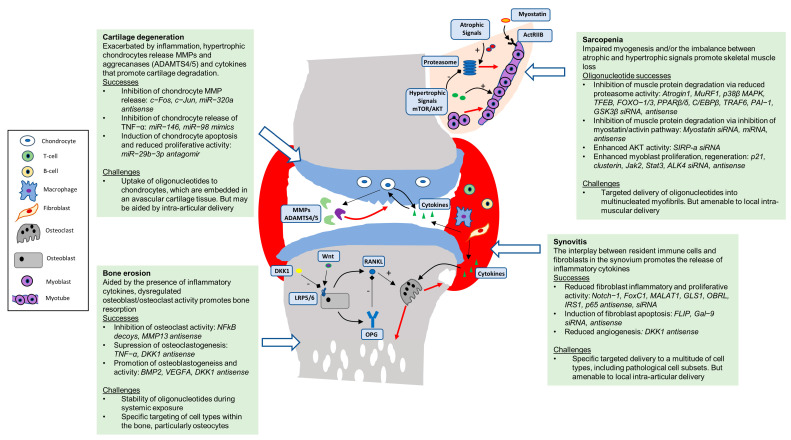
Oligonucleotides targeting pathological cellular processes in the tissues of the musculosketal system. ADAMTS 4, 5; a disintegrin and metalloproteinase with thrombospondin motifs 4, 5. BMP; bone morphogenic protein. DKK1; Dickkopf-elated protein 1. FOXC1; forkhead box C1. MMP; matrix metalloproteinase. OPG; osteoprotegerin. RANKL; receptor activator of nuclear factor kappa-Β ligand. TNF-α; tumour necrosis factor. Gal-9; Galectin9. MALAT1; Metastasis Associated Lung Adenocarcinoma Transcript 1. GLS1; Glutaminase-1. OBRL; Long leptin receptor. PAI-1; plasminogen activator inhibitor-1. TFEB; transcription factor EB. IRS-1; insulin receptor substrate-1. SIRP-α; signal regulatory protein alpha. C/EBPβ; CCAAT/enhancer-binding protein beta.

## Data Availability

Not applicable.

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
