# Peer review of "Oligonucleotide Therapeutics for Age-Related Musculoskeletal Disorders: Successes and Challenges"

_pharmaceutics, 2023, doi:10.3390/pharmaceutics15010237_

Round 1

Reviewer 1 Report

Manuscript pharmaceutics-2107201 has been carefully reviewed. It is scientifically sound and appropriately organized. However, there are some minor points to be considered before publishing:

- It is highly recommended to add one or two paragraphs as the opening in the "Introduction" section. A standard review article needs to engage the readers with an insightful introduction explaining the main questions and defining the aims and scope of the review study.

- It is suggested to reproduce several schematic figures from the reviewed studies to help the readers in visual understanding.

- In the "Challenges for clinical development" section, it's said that there are multiple oligonucleotide candidates for skeletal muscle targeted therapy under investigation phases. It would be valuable if the authors discuss these successful examples regarding their challenging features.

Author Response

  1. It is highly recommended to add one or two paragraphs as the opening in the "Introduction" section. A standard review article needs to engage the readers with an insightful introduction explaining the main questions and defining the aims and scope of the review study.

The reviewer raises a good point.  In the revised manuscript we have included an introductory paragraph to define the aim and purpose of the review.

  1. It is suggested to reproduce several schematic figures from the reviewed studies to help the readers in visual understanding.

This is a good suggestion. In the revised manuscript we have include a single large schematic, rather than a number of individual schematics given the overlap of pathology and cellular mechanisms across the different musculoskeletal disorders.

  1. In the "Challenges for clinical development" section, it's said that there are multiple oligonucleotide candidates for skeletal muscle targeted therapy under investigation phases. It would be valuable if the authors discuss these successful examples regarding their challenging features.

Apologies for the lack of clarity here.  The challenges of all these investigative oligonucleotide candidates are discussed in the sub-sections of this section. Namely, stability and tissue penetration. We make specific reference here to all of the oligonucleotide candidates designed to modulate targets in the cartilage, since as an avascular tissue, penetration into the chondrocyte cells is a particular challenge.

Reviewer 2 Report

I suggest moving chapter 3 as the introduction. The introduction should then be expanded with an overview of the discussed modifications (if possible with a picture of these chemical modifications) and also with an explanation of the principle of action of oligonucleotides discussed later in the text (recruitment of RNase H, splicing modulating oligonucleotides, siRNA, etc.).

Chapters 1 and 2 could be combined and divided into the subsections used in Chapter 2.

Consider redesigning the table at the end of the article into a format that would fit on one page and moving it to the introduction.

Author Response

  1. I suggest moving chapter 3 as the introduction. The introduction should then be expanded with an overview of the discussed modifications (if possible with a picture of these chemical modifications) and also with an explanation of the principle of action of oligonucleotides discussed later in the text (recruitment of RNase H, splicing modulating oligonucleotides, siRNA, etc.).

Thank you for this suggestion. As suggested in the revised manuscript we have expanded the introduction. However, we felt it was based to keep the current chapter 3 as chapter 3. This is because chapter 3 discusses the challenges in relation to specific issues around tissue penetration. We felt therefore it was best to discuss the musculoskeletal disorder pathologies and the different tissues before reflecting on the challenges that remain to be overcome.

  1. Chapters 1 and 2 could be combined and divided into the subsections used in Chapter 2.

Thank you for this suggestion, which we have considered. However, Chapter 1 discusses the pathology and key cellular mechanisms that drive the pathology across different musculoskeletal disorders. For chapter 2, we then evaluate the successes of oligonucleotide-based therapeutics in turn each of the relevant tissue types. To combine these two chapters we feel will make for an overlong chapter and will lose clarity as a result.

  1. Consider redesigning the table at the end of the article into a format that would fit on one page and moving it to the introduction.

We believe it is important and informative that the table provides information on tissue type, oligonucleotide modality, in vitro findings and where applicable information on in vivo administration and dosage. It is difficult to design this to fit onto a single page without making the font size very small. This might be best handled by the journal editing/formatting team.

Reviewer 3 Report

The authors present an original review of oligonucleotide-based therapies for age-related musculoskeletal disorders. The document is well-structured and written. All preliminary notations are also provided. The topic covered is in perfect line with the purpose of the journal, so I suggest the publication of this work. 

Author Response

The authors present an original review of oligonucleotide-based therapies for age-related musculoskeletal disorders. The document is well-structured and written. All preliminary notations are also provided. The topic covered is in perfect line with the purpose of the journal, so I suggest the publication of this work. 

Thank you for taking the time to review our article and for the positive comments.